# Vaccination Therapy for Acute Myeloid Leukemia: Where Do We Stand?

**DOI:** 10.3390/cancers14122994

**Published:** 2022-06-17

**Authors:** Kordelia Barbullushi, Nicolò Rampi, Fabio Serpenti, Mariarita Sciumè, Sonia Fabris, Pasquale De Roberto, Nicola Stefano Fracchiolla

**Affiliations:** 1Hematology & BMT Unit, Fondazione IRCCS Ca’ Granda Policlinico Ospedale Maggiore di Milano, 20122 Milan, Italy; kordelia.barbullushi@unimi.it (K.B.); nicolo.rampi@unimi.it (N.R.); fabio.serpenti@unimi.it (F.S.); mariarita.sciume@policlinico.mi.it (M.S.); sonia.fabris@policlinico.mi.it (S.F.); pasquale.deroberto@policlinico.mi.it (P.D.R.); 2Department of Oncology and Onco-Hematology, University of Milan, 20122 Milan, Italy

**Keywords:** acute myeloid leukemia, vaccination, immunotherapy, dendritic cells

## Abstract

**Simple Summary:**

Immunotherapy is changing the therapeutic landscape of many hematologic diseases. Nevertheless, in acute myeloid leukemia (AML) the anti CD33 antibody-drug conjugate gemtuzumab ozogamicin is the only approved drug. In this review, we aimed at reporting biological mechanisms and their clinical impact of vaccines in AML. The principal vaccination strategies have been analyzed and commented, highlighting advantages in terms of toxicity and possibility to apply in elderly patients. Nevertheless, the clinical results of this strategy in AML are still far from satisfactory. It is necessary to evaluate the best scenario for this approach, whether in a therapeutic, prophylactic, or preemptive setting, considering the poorer results in active or high-burden disease. Finally, we underlined the necessity in AML of further research to optimize immunotherapy-based strategies, among which vaccines might represent relevant actors to contribute to long-term disease control.

**Abstract:**

Immunotherapy is changing the therapeutic landscape of many hematologic diseases, with immune checkpoint inhibitors, bispecific antibodies, and CAR-T therapies being its greatest expression. Unfortunately, immunotherapy in acute myeloid leukemia (AML) has given less brilliant results up to now, and the only approved drug is the antiCD33 antibody-drug conjugate gemtuzumab ozogamicin. A promising field of research in AML therapy relies on anti-leukemic vaccination to induce remission or prevent disease relapse. In this review, we analyze recent evidence on AML vaccines and their biological mechanisms. The principal proteins that have been exploited for vaccination strategies and have reached clinical experimental phases are Wilm’s tumor 1, proteinase 3, and RHAMM. the majority of data deals with WT1-base vaccines, given also the high expression and mutation rates of WT1 in AML cells. Stimulators of immune responses such as TLR7 agonist and interleukin-2 have also proven anti-leukemic activity both in vivo and in vitro. Lastly, cellular vaccines mainly based on autologous or allogeneic off-the-shelf dendritic cell-based vaccines showed positive results in terms of T-cell response and safety, also in elderly patients. Compared to other immunotherapeutic strategies, anti-AML vaccines have the advantage of being a less toxic and a more manageable approach, applicable also to elderly patients with poorer performance status, and may be used in combination with currently available therapies. As for the best scenario in which to use vaccination, whether in a therapeutic, prophylactic, or preemptive setting, further studies are needed, but available evidence points to poorer results in the presence of active or high-burden disease. Given the poor prognosis of relapsed/refractory or high-risk AML, further research is urgently needed to better understand the biological pathways that sustain its pathogenesis. In this setting, research on novel frontiers of immunotherapy-based agents, among which vaccines represent important actors, is warranted to develop new and efficacious strategies to obtain long-term disease control by immune patrolling.

## 1. Introduction

Acute myeloid leukemia (AML) is a clonal disorder of hemopoietic cells characterized by the uncontrolled proliferation of neoplastic blast cells conditioning concomitant bone marrow failure [1]. As a consequence of recent insights into molecular pathways, novel agents have been studied besides traditional chemotherapy in order to achieve and maintain an even deeper complete remission, possibly at the minimal detectable residual disease. Accordingly, drugs such as BCL-2 antagonist (Venetoclax) [2], hypomethylating agents (e.g., Azacitidine [2], Decitabine [3], and targeted therapies (e.g., Midostaurin [4]) have been licensed by the FDA and EMA for the treatment of AML. In this setting, AML relapse still represents the main challenge for physicians, due to the biological evolution of leukemic cells into more aggressive forms and the consequent selection of resistant clones. Integrating molecular, cytogenetic, and intrinsic clinical features (e.g., development from a preexisting myeloid disorder or peripheral high blast counts), three prognostically distinct risk categories (favorable, intermediate, and adverse) were identified in the 2017 revised European Leukemia Net (ELN) recommendations [1].

Among several mechanisms that could be exploited to reduce relapse risk, the immune control of leukemic myeloid proliferation is the base of different therapeutic strategies. In this field, the forerunner approaches are represented by hemopoietic stem cell transplantation (HSCT) and donor lymphocyte infusion (DLI), as efficient graft versus leukemia (GvL) shows that immunological activity can provide a long-lasting disease control. Although effective, HSCT is not applicable to every patient for many possible reasons (unfitness, lack of an available donor, excessive toxicity profile). Consequently, further immunological strategies are being explored: monoclonal antibodies (immunoconjugates or not, checkpoint inhibitors), bispecific antibodies (BiTE), chimeric antigen-receptor (CAR) T cells, and, lastly, vaccination. 

In this review, we discuss the state of the art of vaccination strategies aimed to enhance anti-leukemia immunity and consequential clinical outcomes in AML. 

## 2. Peptide Vaccines

Vaccines in AML could be macroscopically categorized into two broad groups based on the type of constructs: peptide vaccines and dendritic cell (DC)-based vaccines (Figure 1). In both cases, vaccine therapy aims to induce cellular and/or humoral immune response (IR) against specific leukemic antigens. To elicit an effective IR and achieve a valid clinical response, tumor-associated antigens (TAAs) should be highly expressed, highly immunogenic, and restricted mainly to myeloid blasts. A method to evaluate the immune response to a peptide vaccine is represented by tetramer or pentamer staining. In this case, specific T cells activated by a peptide vaccine may show high affinity to the tetramers or pentamers of peptide–MHC complexes compared to monomeric peptide–MHC complexes, enhancing T-cell response quantification sensitivity by immunostaining [5]. Furthermore, the immune response may be detected by studying cytokine quantification either measured by flow cytometry intracellular cytokine staining (ICS) in the endoplasmic reticulum after cell stimulation, or by enzyme-linked immunospot (ELISPOT) assay, measuring cytokines immediately after their secretion [6,7].

The main leukemic antigens investigated are Wilms’ tumor 1 (WT1), proteinase 3 (PR3), the receptor for hyaluronic acid-mediated motility (RHAMM), and mucin 1 protein (MUC1), all able to elicit a specific cytotoxic T response [8].

WT1 is a transcription factor involved mainly in the transcription of growth factors and other genes such as *c-myc* and *bcl-2*. Mutations of WT1 may lead to the dysregulation of cell growth and differentiation, playing a role in leukemogenesis [9]. Humoral and cytotoxic responses to WT1 protein have been described in patients affected by hematological diseases, including AML [10], giving the rationale for its use in vaccine therapy.

Several phase I trials showed the safety and tolerability of WT1 vaccines in AML patients in different clinical settings [11,12,13]. Reported side effects are usually represented by the delayed hypersensitivity reactions of the minimal entity. As for clinical efficacy, most studies focused on relapse prevention in high-risk disease in complete remission (CR). Long-term survival (more than 8 years) was reported in three out of eight patients with high-risk disease (two relapsed, one secondary AML) with minimal residual disease persistence measured through WT1 RNA in bone marrow (BM) or peripheral blood (PB) [14]. Though this is encouraging, appropriately powered clinical trials are needed to further investigate the role of vaccination in this setting. 

A phase II clinical trial evaluated the immune and clinical outcome of an HLA-A*0201-restricted WT1 epitope peptide vaccine administered with granulocyte colony-stimulating factor (G-CSF) in 17 refractory AML patients [15]. An immunological response rate of 44% was described and a significant increase in WT1-specific tetramer responses in peripheral blood (PB) was detected in patients with <40% blasts in BM. Nevertheless, no correlation between immunological response and clinical benefit was shown: stable disease (SD) lasting at least 8 weeks was observed in 10 patients, and only one CR was observed in a patient in partial remission (PR). A transient increase in blast counts was seen in some patients (including the CR patient), suggesting that specific immune responses may require an adequate interval to be mounted [15]. The limits of the study rely on a sample size that is too small to draw clinical conclusions, especially in patients with active disease, where the immunological potential of vaccination may barely be applied.

OCV-501 is a different HLA class II-restricted peptide vaccine derived from WT1, investigated in a phase II, double-arm, double-blind, placebo-controlled trial in CR1 AML patients aged 60 years or older, not eligible for HSCT: 133 patients were randomized, 68 in the vaccine arm and 65 in the placebo arm. The trial did not show any significant immunological response, without any difference between the two arms for disease-free survival (DFS) and overall survival (OS) [16].

In the attempt to enhance immunological responses, mixtures of peptides containing both class I and class II have been proposed to elicit both CD4+ and CD8+ T cell stimulation and sustained response of cytotoxic memory T cells.

In a phase I trial, Brayer and colleagues [11] showed a mean DFS of 244 days (range, 30–445 days) and a mean OS of 608 days (range, 201–1071 days) in 14 AML patients in CR1 and CR2 treated with an oligopeptide vaccine composed of a mixture of peptides including both class I (restricted to HLA-A*0201) and class II epitopes. At the time of analysis, all patients showed disease progression, with three patients alive. Despite the specific strategy directed both to CD8+ and CD4+ stimulation, no consistent and measurable WT1-specific T-cell response was detected. Of note, the authors highlighted that the use of WT1 class I epitopes restricted to HLA-A*0201 in a population not selected for HLA-A expression could have limited CD8+ response [11]. This may suggest the need for a specific HLA selection of patients suitable for treatment with peptide vaccines, which, however, can consistently reduce the percentage of patients that could be treated with this strategy.

Double CD8+ and CD4+ stimulation was evaluated also in a phase I/II trial [17]. A vaccine made of two distinct WT1 HLA-A2-restricted epitopes and HLA-DR T helper cell epitope (PADRE) was administered in 8 HLA-A*0201-positive patients with poor-risk AML at different stages of disease. Early response to vaccination was observed in 86% of patients as assessed by tetramer and functionally by ELISPOT assays. However, a functional WT1-specific immunological memory was not documented despite the double stimulation. Since significant clinical efficacy was lacking, it was supposed that adequate CTL memory is necessary for the vaccine to be effective [17].

More promising results emerged from the study on Galinpepimut-S (GPS) [18]. GPS is a multivalent vaccine consisting of class I designed for HLA-A*02:01 and class II-restricted peptides from WT1. It was studied in a phase II study, in which 22 patients with AML (of whom nine patients had HLA-A*02:01) in CR1 were treated with GPS, administered together with G-CSF at least for six vaccinations, with the possibility of six additional monthly doses in patients remaining in CR1. The 3-year OS was 47.4% and the median DFS from CR1 was 16.9 months (being 8 months the median time in CR1 prior to the first GPS administration), with the median OS not reached. Relapse occurred in 15 patients (68%). As for IR, nine patients were tested for CD4+ response, and seven of nine HLA-A*02:01 patients were tested for CD8+ response. Overall IR was confirmed in 64% of patients, with rates of 44% and 86% of specific CD4+ and CD8+ responses, respectively. Survival curves were compared between immunological responders and non-responders, showing a trend for better DFS and OS in the formers [18]. Even if these results looked promising, a randomized trial is needed, to confirm the clinical efficacy of GPS treatment.

Another antigen widely investigated in vaccine engineering is RHAMM, overexpressed in nearly 80% of AML cases [19]. In a phase I trial, 10 patients (three with AML, three with myelodysplastic syndrome, and four with multiple myeloma) with limited tumor burden were treated with an R3-based vaccine, a RHAMM-derived epitope peptide for CD8 T cells. Among the three leukemic patients, only one showed an IR at tetramer staining, ELISA (increase in IL2 levels), and ELISPOT (increase in INFγ levels), experiencing blast reductions in the bone marrow, but subsequent relapse after 23 months [19].

Combined vaccine approaches have been also investigated, involving PR3, a TAA highly expressed by AML blasts, which plays a role in the deregulation of the NFκB pathway in AML [20]. Kuball and colleagues [8] investigated a combination of PADRE, adjuvants, and either WT1 or PR3 vaccines in AML patients. Unexpectedly, in this study, neither clinical nor immune response was obtained. Moreover, WT1-specific T cells observed in some patients before vaccination were no longer detected after that. These results raise an important issue with vaccination therapy and its immunogenicity. In fact, depending on antigen and adjuvants used, vaccination may paradoxically lead to immune exhaustion, anergy, or the deletion of reactive T cells. Of note, Rezvani and colleagues showed the deletion of high-avidity CD8+ T-cell response caused by repeated vaccine injections [21]. Thus, to elicit CD8+-specific cell proliferation sustained by adequate T-helper activity, parameters to consider to obtain adequate vaccine immunogenicity are targeted antigen, adjuvants, GM-CSF co-stimulation, and the number of booster injections. Kuball et al. speculated that the failure of their vaccine could be related to the adjuvant used (MontanideISA51) [8] which should raise caution for further construct compositions. 

In terms of the future development of peptide vaccines, the experiences reported suggest many considerations.

As far as vaccine engineering is concerned, the main points to be taken into account are the selection of appropriate TAA, immunogenic adjuvants, and the selection of HLA-compatible patients when HLA-restricted epitopes are used. From a clinical point of view, some interesting fields may be the post-transplant vaccination, the ex vivo expansion of a specific T-cell subset to boost and equilibrate immune response, and the design of rational association strategies with other immune therapies, such as checkpoint inhibitors (in order to avoid the emergence of a tolerogenic milieu). As for the last point, a phase 1/2 trial evaluating the efficacy and safety of combined treatment with GPS and the programmed death-1 (PD1) inhibitor pembrolizumab in AML and other solid cancers is now recruiting (NCT03761914).

When considering clinical trial design, the most promising strategies should be tested in randomized studies, probably in the setting of patients in CR or with MRD persistence.

## 3. DC Vaccines

Dendritic cells play a major role in the immune system by stimulating both adaptive and innate immune responses. Via major histocompatibility complex (MHC) class I and II molecules, DCs can present the antigens both to CD8+ cytotoxic T-lymphocytes (CTLs) and CD4+ helper T cells that can differentiate into T-helper type 1 (TH1) cells and activate CTCLs-mediated cell lysis [22]. At the same time, DCs can induce innate immune responses by recruiting and stimulating natural killer cells, mainly by cytokines such as IL12 and IL-15 [23]. Based on these observations, DC-based vaccines have been explored to trigger antileukemic responses and overcome immunologic escape.

DCs could originate from different progenitors, but the most widely studied for vaccine therapy are those derived from leukemic cells (so-called AML-DCs) [24] and monocytes (mo-DCs) [25] (Figure 1). When comparing the two cell lines, AML-DCs may have the advantage to present, along with MHC-class molecules, directly tumor-associated antigens (TAAs) on the cell membrane. Contrarily, mo-DCs need to be loaded with TAAs, either by incorporating messenger RNA (mRNA) by electroporation [26,27,28] or pulsing dendritic cells with apoptotic AML cells, their lysates [29,30] or directly with peptides themselves (e.g., WT1). Furthermore, evidence about the presumptive role of AML-DCs in immune tolerance [31,32] raised doubts about their real efficacy. In a direct comparison between the performance of mo-AML-DCs and DCs [33], the latter have proven to be more effective in activating autologous leukemia-specific T cells, showing a higher immune response, supposed to be related to the expression of the co-stimulatory molecule 4-IBB. 

Most experiences with DC-based vaccine therapy have been reported in the setting of minimal residual disease (MRD) in order to prevent leukemia relapse, especially in patients considered unfit for more intensive therapies.

Similar to peptide vaccines, WT1, alone or in combination with other TAAs, represents one of the most-studied antigens. Anguille et al. [34] explored the efficacy of a vaccine based on autologous DCs loaded with three different constructs of WT1 in patients in first CR at high risk of relapse after chemotherapy, as defined by prior MDS/MPN, the persistence of WT1 mRNA, or FLT3-ITD mutation. When compared to a similar population from the Swedish Registry, a higher five-year OS was reported in the vaccinated group compared to controls, with no significant difference in relapse-free survival, although longer in responder vs. non-responders (5-year RFS 50% vs. 7.7%). Better outcomes were reported in patients younger than 65 years who did not receive HSCT. Hereby, a possible favorable synergic effect of vaccines on subsequent chemotherapy was supposed in the case of relapse, but the small sample size did not allow for the drawing of definite conclusions. Moreover, in 69% of responding patients MRD negativity was obtained, determined by WT1 levels and other molecular markers, suggesting the possibility of an effective control of minimal leukemic persistence. Dagvadori et al. [35] offered an alternative approach to WT1-based DC vaccination with fusion proteins consisting of an antibody specific for the DEC205 endocytic receptor on human DCs and different fragments of WT1. Preliminary ex vivo assays identified the anti-hDEC205-WT1 91–138 chimeric protein as the only construct with a significant CD4+ and CD8+ co-stimulatory effect, independently of HLA-A restriction. Currently, no experiences in vivo have been reported. In another phase I/II trial [36], autologous DCs loaded with WT1 and PRAME (Preferentially Expressed Antigen of Melanoma), another antigen expressed on leukemic cells, were used to vaccinate 20 AML patients in CR considered unfit for transplantation. CR was maintained in 60% of patients with a short follow-up of 12 months. Immunologic T-cell responses to WT1 and/or PRAME tested by IFN-γ ELISPOT were detected more frequently in the peripheral blood of patients experiencing a relapse than those maintaining CR. This finding raises some questions regarding T-cell response, which seems on one side to be dependent on the presence of leukemic blasts, on the other not efficient enough to kill them.

Promising results using next-generation DCs loaded with WT1, PRAME, and CMVpp65 and cultured with TLR agonists were obtained in 10 AML patients in first CR with the non-favorable risk group or MRD positivity [27]. Half of the patients were in CR after a median observation of 1057 days from diagnosis and 811 days from the first vaccination, with the best responses observed in patients younger than 65 years old. Moreover, in this study, although promising, clinical results should be considered cautiously because of the limited patient numbers in a phase I setting and should be confirmed in appropriately designed clinical trials.

In order to augment T-cell activation, the group of Kitawaki et al. [37] described the use of pulsed DCs with modified or natural WT1 and zoledronate, exploiting the bisphosphonate property of activating antigen-specific CD8+ T cells. The study was conducted on three AML patients. A rapid decline of T-cells in peripheral blood was registered after vaccination with modified WT1 peptide-loaded DCs. Contrarily, T-cell response was maintained after 3 months in patients vaccinated with DCs loaded with natural peptide WT1. This evidence suggests caution in engineering DCs with modified antigens.

Chevallier et al. [29] explored the possibility of preventing relapse in a phase I/II trial. They consolidated CR with autologous mo-DCs pulsed with irradiated autologous apoptotic blasts. Among twenty-one elderly patients (range 65–84 years), only five achieved CR and were then vaccinated. Despite the small size of the study, almost all patients had adverse genetic risk profiles according to ELN2017 classification (two secondary AML and two FLT3 positive AML). Nevertheless, the reported median OS in this small group of CR patients was longer than the historical data (16 months vs. 8 months), with 60% maintaining MRD negativity. Similarly, Rosenblatt et al. [38] supported the potential efficacy of a vaccine realized by fusing AML cells with autologous DCs in AML patients. In a sample with almost half of patients with intermediate or high-risk AML, a 4-year PFS rate of 71% was documented, with a good tolerability profile and low incidence of adverse events.

Allogeneic compared to autologous DCs may enhance T cell immune activity by stronger Th1 cell differentiation and activation in response to alloantigens [39]. On the basis of this observation Van de Loosdrecht et al. [30] used allogeneic DC vaccines expressing multiple TAAs (e.g., WT1, PRAME) and MHC class I/II molecules in 12 elderly patients either in CR1/2 or in relapse with stable blast counts. At the end of the study, disease-free status was reached in one and maintained in four patients. A longer OS, possibly associated with a more pronounced T-cellular response, was reported in patients with no circulating blasts at treatment. Interestingly, in two out of three patients evaluated, antibodies against autologous blasts were detected. In the recent update of a phase II trial (Loosdrecht et al., Abstract ASH 1274) [40], Loosdrecht et al. reported how allogeneic AML-DCs may be effective in eradicating minimal residual disease documenting MRD conversion in four patients among 19 treated.

Another relevant target in vaccine therapy is telomerase, since it is significantly expressed in patients with AML and has a role in maintaining leukemic stem cell compartment, in particular in high-risk cytogenetics. Khoury et al. [28] transfected mature DCs with mRNA-encoding telomerase (human telomerase reverse transcriptase (hTERT)) and the lysosomal-associated membrane protein (LAMP), a lysosomal targeting signal, to enhance immunoreactivity. Twenty-two intermediate or high-risk AML patients were vaccinated with a median number of 19 hTERT-DC infusions: 58% of them maintained CR after a median follow-up of 52 months.

The efficacy of vaccine therapy has also been tested in small series of patients with active disease. Massumoto et al. [41] reported the case of a female patient, who relapsed after two lines of therapy and was considered unfit for HSCT, who received a hybrid vaccine obtained by fusing allogeneic DCs and autologous leukemic cells and documented stable disease for 9 months. Similarly, Li et al. [42] described in a group of five patients a transient stabilization of leukocyte counts in two of them with a partial reduction in peripheral blasts after autologous AML-DC vaccines. Moreover, Kitawaki et al. [43] documented stable disease as the best response in two of four patients treated with dendritic cells pulsed with autologous apoptotic leukemic cells. These studies represent biological proof of the concept of immunological activity even in active disease but confirm, at the same time, their limited efficacy in this setting.

In a prospective case-control study, Dong M et al. [44] treated newly diagnosed AML elderly patients with low-dose chemotherapy with or without autologous DCs and cytokine-induced killer (CIK) cells (21 patients vs. 23 patients). A higher overall response rate in the immunochemotherapy group (ORR 71% with 29% in CR vs. ORR 39% with 22% in CR) was registered. In patients receiving DCs, they described a significant increase in Th1 response cytokines and a higher percentage of T cells in peripheral blood, regarding this approach as feasible and effective in case of minimal disease burden.

DC vaccines have been evaluated also after HSCT to induce anti-leukemic specific responses in the donor immune system to prevent relapse [45]. In this setting, a clinical trial was conducted in 48 patients affected by AML and acute lymphoblastic leukemia (ALL), comparing DC vaccines in nineteen AMLs and four ALLs to DLI infusions in twenty-two AMLs and three ALLs. DCs were transfected with an adenovirus to deliver two TAAs (survivin and MUC1) and flagellin, a TLR5 agonist acting as a potent immunological adjuvant. In addition, Suppressor Of Cytokine Signaling 1 (SOCS1) was silenced into autologous mo-DCs [46]. SOCS1 is an intracellular immune checkpoint whose inhibition enhances the immunopotency of DCs and contributes to overcoming self-tolerance. Its silencing was obtained through short-hairpin RNA. The treatment with the vaccine was followed by the infusion of cytokine-induced killer cells (CIKs). Higher OS was also obtained in the vaccine compared to the DLI group: 48.9% versus 27.5% at 3 years (*p* = 0.028). No grade 3–4 aGVHD and cGVHD were described in the vaccine group versus 36% in the DLI group (*p* = 0.001).

As vaccine therapy was followed by the infusion of CIKs, the role of each therapy on clinical outcomes is hard to determine. Nevertheless, this interesting sequential approach with different immunological strategies may represent a promising new therapeutic strategy to enhance the immunological control of disease.

Other smaller studies showed promising safety and immune responses using various vaccine constructs in the post HSCT setting: WT1- and keyhole limpet hemocyanin (KLH)-pulsed donor-derived mo-DCs vaccine [37] and WT1-loaded donor-derived mo-DCs vaccine infused in association with DLI [45]. Clinical outcomes, however, did not prove an advantage for vaccine therapy.

Another aspect to be considered is the time required for vaccine production (median time from leukapheresis to first vaccination is 25 days): some patients may progress in the meantime, especially in case of residual disease, becoming ineligible for vaccination or limiting its efficacy.

The therapeutic role of DCs vaccine in the post-HSCT setting is a field that deserves further investigation, in particular selecting the proper strategy to engineer DCs and design combined strategies with different immunotherapies.

Table 1 summarizes the major reported trials with vaccine therapy in AML.

## 4. Comparison to Other Immunotherapies

Other anti-AML immunotherapies include cell-targeting antibody–toxin conjugates, immune checkpoint inhibitors, bispecific antibodies, and CAR-T cell-based strategies (Figure 2) [47]. Overall, these strategies are giving promising results in terms of disease response, higher than those reported for vaccination therapies, even if associated with a certain degree of toxicity.

Antibody-based therapies are directed to cell surface antigens which limit their applicability with respect to other neoplasms, such as lymphoid ones, due to the difficulty in identifying robust cell surface targets expressed selectively on neoplastic cells but not on their normal counterpart. To date, as for cell-targeting antibodies, only two toxin-conjugate antibodies are approved for AML therapy, namely gemtuzumab ozogamicin (Mylotarg) for AML and tagraxofusp for plasmacytoid dendritic cell neoplasms. Both agents have been proven to significantly increase event-free survival when added to chemotherapy (for Mylotarg, ALFA0701 [48]) or as a single agent (for tagraxofusp, STML 401 [49]) but at a non-negligible toxicity cost. Mylotarg is associated with a high incidence of gastrointestinal side effects, infusion reactions, and a rate that can reach 20% of veno-occlusive disease after HSCT. Tagraxofusp has been linked to infusion reactions and capillary leak syndrome, also with fatal outcomes.

The most-tested immune checkpoint inhibitors in AML are anti-PD-1 antibodies. However, responses are variable and often transient, as PD1-L is not constitutively expressed on all neoplastic myeloid cells [50]. PD1-L expression can be increased with hypomethylating agents [51], thus making combination therapy with them promising. Nowadays, toxicity related to immune checkpoint inhibitors is manageable, with the exception of the increased risk of graft-versus-host disease in the case of previous or subsequent HSCT [52].

Bispecific antibodies are engineered drugs recognizing two different antigens in order to recruit an immune cell (mainly T cell or NK cell) and link it to the neoplastic cell. Antibodies currently under study are directed against different key antigens, mainly CD33, CD123, and CLEC12A/CLL-1 [53,54]. Effectiveness data are still not available, as most clinical trials are ongoing. As known for other settings, bispecific antibodies bear the risk of neurotoxicity and cytokine release syndrome, beyond persistent cytopenia and liver toxicity [55].

CAR-T cells are engineered autologous lymphocytes with a polypeptide chain made by the combination of the extracellular antigen-binding site of an antibody and the intracellular CD3ζ chain linked to the signal cascade of the T cell receptor, thus bypassing the need for an HLA-based immune response mechanism, as HLA downregulation is a common way of resistance for leukemic cells. Given the good results of CAR-T cell therapy in lymphoid neoplasms [56,57], there is great interest in CAR-T cell therapy also in AML, with, however, similar difficulties as with other antibody-based approaches, due to the difficulty in identifying the proper target antigen [58]. Several trials are ongoing with different CAR-T structures, but up-to-date reported experience has not described long-term disease control uniquely with CAR-T infusion. CAR-T cell toxicity, similar to but even more strongly than bispecific antibodies, lies in cytokine release syndrome and neurotoxicity (especially in the first weeks after infusion), prolonged cytopenia, and liver toxicity.

If, in the context of active disease, other immunotherapies show greater efficacy, more has to be studied in different clinical settings, such as relapse prevention and MRD positivity, in which vaccines may provide a maintenance strategy at low risk of toxicities. Moreover, an interesting open question can be made on the use of vaccines in pre-leukemic states (namely lower risk myelodysplasia, or also clonal hematopoiesis of indeterminate potential (CHIP) [59]), in which aggressive and more toxic therapies are not acceptable. Research on pre-leukemic states and how to select patients at high risk for leukemia development has to proceed in parallel to vaccine research with the aim of leukemia prevention.

As for other immunotherapies, with the exception of bispecific antibodies and CAR-T, vaccination therapy shares the feature of being an easily combinable agent with standard approved therapy in order to increase response rates and response duration. Nevertheless, the possibility that vaccination may induce tolerance in the effector T cell deserves further investigation, and the association in sequence with immune therapies that rely on these cells as bispecific antibodies and checkpoint inhibitors should be carefully evaluated. Overall, AML remains a proliferating tumor with a high tendency for immune escape, leading to the possible failure of these immunotherapies.

Also, differently from CAR-T cell therapy, peptide vaccines can be readily available drugs that can be rapidly administered to patients, while CAR-T cells require weeks for production, during which AML patients can commonly face complications or die from rapidly progressive disease.

Lastly, to a different extent, all cited immunotherapies bear a significant economic cost which can constitute a limit in the case of the wide application of these therapies or for their application in developing countries.

## 5. Conclusions

Nowadays, relapsed/refractory or high-risk AML is still a poor-prognosis disease, with unsatisfying outcomes. New therapeutic approaches are needed to reach deeper and long-lasting disease control and prevent relapse. Immunotherapy offers a good strategy for this purpose and the immune control of leukemic cells can be achieved with different mechanisms and agents. Vaccine-based strategies have been explored in clinical trials, several still ongoing (Table 2). Surely, the safety and the feasibility of vaccines is a major point of interest for their use, especially in older patients, but relevant immunological and clinical outcomes are missing, with few exceptions so far. In particular, results in active disease are poor, and overall, the best setting for the application for vaccines seems to be the post-chemotherapy and post-alloHSCT CR setting as a maintenance therapy to reduce the risk of relapse. An open question remains as to the possibility to use vaccines as preventive strategies in pre-leukemic states, such as lower-risk myelodysplasia or CHIPs [59]. In order to improve immune control and in the end achieve better clinical outcomes, vaccines may be associated with further immunological strategies, i.e., immune checkpoint and macrophage checkpoint blockade. Further research is needed to better define immunotherapy-based schemes of treatment, in which including vaccines possibly significantly improves the therapeutic scenario of AML.

## Figures and Tables

**Figure 1 cancers-14-02994-f001:**
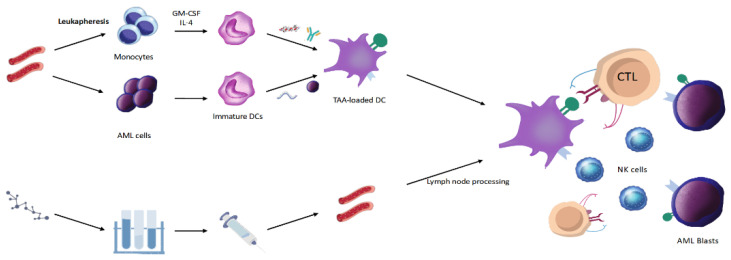
Main mechanisms for vaccine-based immunotherapy production and function in acute myeloid leukemia (AML). Top left: CD14+ monocytes or blast cells are collected from peripheral blood and differentiated into immature dendritic cells (DCs) in culture with IL-4 and GM-CSF. Mature DCs loaded with tumor-associated antigens (TAA) are then obtained by specific procedures (e.g., mRNA electroporation, pulsing with apoptotic AML cells or their lysates, or directly with peptides). Mature TAA-loaded DCs are reinfused in patients. Bottom left: peptide vaccine production. Preselected peptides (e.g., WT1, RHAMM) are prepared in a laboratory as modified constructs or integrated with other molecules and reinfused in patients. In lymph nodes, they are internalized, processed, and exposed by antigen-presenting cells (APC), particularly Ds. On the right, the terminal stage of vaccine therapy is represented, in which DCs induce the activation of immune effectors CD8+ cytotoxic lymphocytes (CTLs) and NK-cells, which attack, in turn, AML blasts.

**Figure 2 cancers-14-02994-f002:**
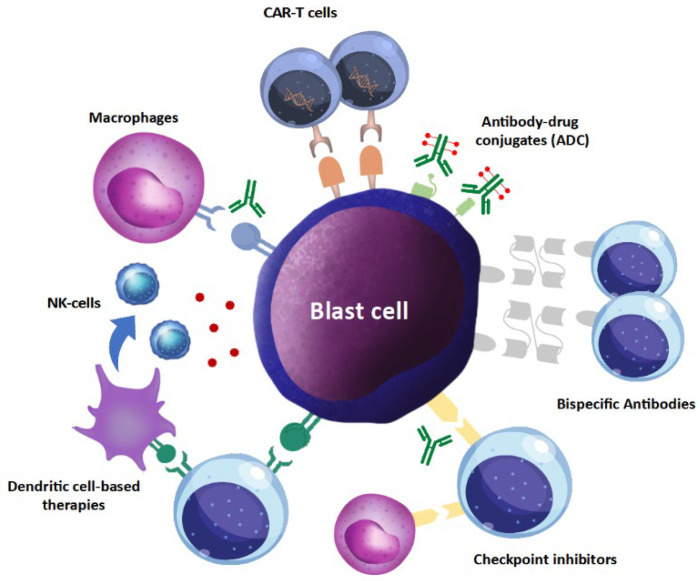
Overview of the immune-based therapeutic approaches under investigation. Starting from the top right and going clockwise: ADC are monoclonal antibodies against a specific tumor antigen, linked to a drug/toxin which is internalized in the target cell, causing its death. Bispecific antibodies have two targets, one specific for the tumor cell and the other for T cells, and function as T-cell engagers. Checkpoint inhibitors enhance anti-leukemic immune responses by the inhibition of the suppressive effect of checkpoint proteins. Dendritic cell-based therapies consist of vaccination strategies either with peptides or directly with DCs loaded with tumor antigens which are then presented on their surface to both T and NK cells. A macrophage-based therapy is represented by magrolimab, an anti-CD47 monoclonal antibody that interferes with the suppressive signals for macrophages, working similarly to checkpoint inhibitors. Finally, there are cellular therapies with adoptive NK cell infusions or CAR-T cells. Chimeric antigen receptors are surface molecules with an antigen-recognition domain linked to signaling regions of the T-cell receptor pathway, capable of T-cell activation.

**Table 1 cancers-14-02994-t001:** Summary of the most relevant clinical trials available on vaccination therapy in AML. Ref, reference. P, peptide. DC, dendritic cell. Mo, monocyte. Allo, allogeneic. CIK, cytokine-induced killer cells. CT, chemotherapy. GCSF, granulocyte-colony stimulating factor. AML, acute myeloid leukemia. CR, complete response. R/R, relapsed/refractory. HSCT, hematopoietic stem cell transplant. N, number. PB, peripheral blood. DLI, donor lymphocyte infusions. Mol, molecular. PR, partial response. SD, stable disease. DFS, disease-free survival. MM, multiple myeloma. ORR, overall response. M, median. OS, overall survival. FU, follow-up.

Refs.	Vaccine Type	Vaccine Antigen	Clinical Setting	N of pts	Doses	Endpoint	Results
Brayer 2015 [11]	P	WT1	CR high risk AML	16	12	Phase I—safety	No toxicity
Maslak 2010 [12]	P	WT1	CR AML, WT1 +	9	6	Phase I—Immune response	7 out of 8 evaluable pts
Keilholz 2009 [15]	P	WT1 (+GCSF)	R/R AML	17	Median 11	Response rate	4 PR, 1 CR, 3 SD
Uttenthal 2014 [17]	P	WT1	R/R AML	8	8	Response rate and immune response	No correlation
Maslak 2018 [18]	P	WT1	CR1 AML	22	12	Clinical outcome	DFS 16,9 months
Schmitt 2008 [19]	P	RHAMM	Active AML and MM	10	4	Response rate	3 PR in AML and 2 PR in MM
Kuball 2011 [8]	P	WT1, PR3, PADRE	Active AML and MM	9	4	Response rate and immune response	No clinical or immune response,
Anguille 2017 [34]	DC	WT1	CR AML	30	4, then every 2 mo	Relapse rate	relapse reduction rate of 25%
Eckl 2019 [36]	DC	WT1, PRAME	R/R AML	20	4, then every 6 w	Response rate	SD 60%
Lichtenegger 2020 [27]	DC	WT1, PRAME, CMVpp65	CR high-risk AML	10	10	Immune response	higher in CMVpp65
Chevallier 2021 [29]	moDC	-	CR high-risk AML	5	5	Clinical outcome	prolonged OS vs. historical cohorts (16 vs. 8 mo)
Rosenblatt 2016 [38]	DC	-	CR high-risk AML	17	3	Immune response	leukemic reactive CD8+ T cells
Loosdrecht 2018 [30]	alloDC	WT1, PRAME	CR / SD AML	12	4	Clinical outcome	1 CR, prolonged mOS in CR pts (36 mo)
Khoury 2017 [28]	DC	hTERT, LAMP	CR int-risk AML	22	17	Clinical outcome	DFS 58%—mFU 52 mo
Dong 2012 [44]	DC + (CIK + CT)	-	I line AML (vs. CT alone)	21	1	Clinical outcome	ORR 71 vs. 39% CR 29 vs. 22%
Wang 2018 [46]	DC	MUC1, flagellin, SOCS1	Post-HSCT AML rel	35	1	ph I (+CIK) vs. DLI, ph II (+CIK) for early mol relapse	higher OS vs. DLI (48.9 vs. 27.5%) molecular CR in 83% in ph II

**Table 2 cancers-14-02994-t002:** Ongoing trials about vaccine therapy in AML.

Study	Type	Intervention	Disease Status
NCT01686334	Randomized phase II	Wilms’ Tumor (WT1) Antigen-targeted Dendritic Cell Vaccination	Prevent Relapse in Adult Patients With Acute Myeloid Leukemia with MRD positivity
NCT05000801	Phase I	WT1/hTERT/Survivin-loaded DCs	Prevent relapse in adult AML patients with MRD at very high risk of relapse
NCT03059485	Randomized phase II	Dendritic Cell/AML Fusion vaccine (DC/AML vaccine	Prevent Relapse in Adult Patients With Acute Myeloid Leukemia with no history of allogeneic transplantation
NCT03679650	Non-randomized phase I	Dendritic Cell/AML Fusion vaccine with or without decitabine	Prevent Relapse in Adult Patients With Acute Myeloid Leukemia with no history of allogeneic transplantation
NCT04747002	Randomized phase II	peptide vaccine DSP-7888	Prevent relapse in adult AML patients ineligible for HSCT with MRD at very high risk of relapse
NCT03761914	Phase I/II	Wilms Tumor-1 (WT1)-targeting multivalent heteroclitic peptide (Galinpepimut-S) with Pembrolizumab	Active disease

## Data Availability

Non applicable.

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
