# Peer review of "Vaccination Therapy for Acute Myeloid Leukemia: Where Do We Stand?"

_cancers, 2022, doi:10.3390/cancers14122994_

Round 1

Reviewer 1 Report

Barbullushi et al. have highlighted the advances in vaccine research in AML. The review includes all the studies conducted to develop vaccines for AML. However, the review has mostly just listed all the studies without going too deep into the expert´s point of view on the pros and cons of the study. Also, AML is a disease of genetics and that angle was not at all discusses in relation to vaccine development. For example, could vaccines help in case of preleukemic mutations ? 

The rationale and strong future strategies to improve current trends of vaccine development should be included in this review to make it more informative to the general audience.

Author Response

We thank the reviewer for the kind and useful response and comments. We uploaded our revised version of the review, in which we extensively modified the paper thanks also to the reviewer suggestions. In particular, we included two figures dealing with immune therapy mechanism of action and one more table summarizing the major trials cited throught the review regarding vaccination therapy in AML.

Barbullushi et al. have highlighted the advances in vaccine research in AML. The review includes all the studies conducted to develop vaccines for AML. “

- We thank the reviewer for the kind comment.

However, the review has mostly just listed all the studies without going too deep into the expert´s point of view on the pros and cons of the study.”

- Thank you for the comment, we extensively revides the manuscript cutting some redundant information of the cited studies and adding comments on possible pros and cons of them. Especially the “DC vaccine” part was extensively reviewed and rewritten.

Also, AML is a disease of genetics and that angle was not at all discusses in relation to vaccine development. For example, could vaccines help in case of preleukemic mutations ? “

- We thank the reviewer for the interesting point raised with his/her comment. The aspect of therapy of pre-leukemic states is a hot topic today and an important field of research. Up to now, no report has been made on the use of vaccine therapy on pre leukemic states although, as you suggest, it can be a field of application in the future. We remarked this point in the final part of the manuscript and in the conclusion, pointing to it as an open question to study in the future.

The rationale and strong future strategies to improve current trends of vaccine development should be included in this review to make it more informative to the general audience.”

- We revised extensively the manuscript also paying more attention on future perspectives of the different kinds of immunetherapies available nowadays.

Reviewer 2 Report

In their work, Barbullushi K et al. attempt a critical review of the developing vaccination therapies for AML.

Although an intriguing and innovative subject in general, vaccinations in AML as presented in this particular manuscript lacks both the structure and the content to make an impact in this expanding field and should therefore undergo extensive revisions to meet the journal’s standards. To be more precise:

  • Statements in introduction are not supported by references. None of the information presented in the introduction is supported by a reference.
  • Images-figures that describe the mechanism of action of each vaccine category along with a visual representation of the comparison with other immunotherapy strategies are majorly lacking. This makes the manuscript somewhat stiff and hard to comprehend. I would strongly recommend that the authors add at least 2 figures to visualize the above.
  • Extensive English editing is mandatory. In fact, before even addressing no 1) and no 2), English should be corrected because at this state the manuscript is extremely difficult to comprehend
  • Try updating bibliography into a more extended and recent one.
  • What would you think of sequential chemotherapy and vaccination therapy? Is there any pathophysiological background for such an approach?
  • Why in your opinion, are vaccines, esp. DC based so ineffective in AML?

Author Response

We thank the reviewer for the kind and useful response and comments. We uploaded our revised version of the review, in which we extensively modified the paper thanks also to the reviewer suggestions. In particular, we included two figures dealing with immune therapy mechanism of action and one more table summarizing the major trials cited throught the review regarding vaccination therapy in AML.

In their work, Barbullushi K et al. attempt a critical review of the developing vaccination therapies for AML.

Although an intriguing and innovative subject in general, vaccinations in AML as presented in this particular manuscript lacks both the structure and the content to make an impact in this expanding field and should therefore undergo extensive revisions to meet the journal’s standards.”

- We thank the reviewer for the comment. We extensively revised our manuscript cutting some redundant information of the cited studies and adding comments on possible pros and cons of them. Especially the “DC vaccine” part was extensively reviewed and rewritten. We hope the revised version will be more informative and more structered.

To be more precise:

 Statements in introduction are not supported by references. None of the information presented in the introduction is supported by a reference.”

- Thank you for the comment. We added ELN guidelines as reference for general AML; we go deeper in immune therapy citation throughout the rest of the manuscript.

Images-figures that describe the mechanism of action of each vaccine category along with a visual representation of the comparison with other immunotherapy strategies are majorly lacking. This makes the manuscript somewhat stiff and hard to comprehend. I would strongly recommend that the authors add at least 2 figures to visualize the above.”

- We thank the reviewer for the comment. We acknowledged the lack of images being the the major weak point of our review. We added two figures and one more table to try and be as clear as possible and make the paper easier to read and to understand.

Extensive English editing is mandatory. In fact, before even addressing no 1) and no 2), English should be corrected because at this state the manuscript is extremely difficult to comprehend”

- We extensively revised the manuscript for grammatil errors and mispelling

Try updating bibliography into a more extended and recent one.”

- We tried and reporting the major trials in the field, we included a table summurizing the most important studies cited in the article. There is also a table regarding the major open trial nowadays.

What would you think of sequential chemotherapy and vaccination therapy? Is there any pathophysiological background for such an approach? Why in your opinion, are vaccines, esp. DC based so ineffective in AML?”

- We thank the reviewer for the interesting comment. In our revision we stressed the promising application of vaccination in post CT setting, mainly in CR or in MRD+. We also commented on the inefficacy of vaccine therapy in active AML and on the possible machanism of immune exhaustion or immune escape by AML blasts as mechanisms of resistence to vaccine therapy.

Reviewer 3 Report

This manuscript by Barbullushi et al., discusses advances in research on AML vaccines and their biological mechanisms. Here are my comments:

  1. There are typos and grammatical errors throughout the manuscript. e.g. line 54, 325 and so on. Please correct them.
  2. The manuscript can use some figures and more tables to provide more clarity.
  3. Table 1 is not in MDPI format. Please follow author's guidelines.
  4. The introduction is very short. Please expand and cite PMID:35267471.

Author Response

We thank the reviewer for the kind and useful response and comments. We uploaded our revised version of the review, in which we extensively modified the paper thanks also to the reviewer suggestions. In particular, we included two figures dealing with immune therapy mechanism of action and one more table summarizing the major trials cited throught the review regarding vaccination therapy in AML.

  1. There are typos and grammatical errors throughout the manuscript. e.g. line 54, 325 and so on. Please correct them.

- We extensively revised the manuscript for grammatil errors and mispelling.

  1. The manuscript can use some figures and more tables to provide more clarity.

- Thank you for the comment; we acknowledged the lack of images being the the major weak point of our review. We added two figures and one more table to try and be as clear as possible.

  1. Table 1 is not in MDPI format. Please follow author's guidelines.

- Thank you. We modified the table according the author’s guidelines.

  1. The introduction is very short. Please expand and cite PMID:35267471.

- Thank you for the comment, but we think there is a discrepancy with the article you are suggesting as it deals more with FLT3 inhibition than immunetherapy. Is it correct? For the introduction we cited ELN guidelines as a general introduction to AML.

Round 2

Reviewer 1 Report

The authors have significantly improved the manuscript.

Author Response

We thank the reviewer for the positive comment. 

Reviewer 2 Report

Dear Editor, dear authors,

After following the comments of both myself and reviewer no3 and revising the manuscript accordingly, you have managed to present a well improved version of the original draft.

However, there are still issues to be resolved before the paper is considered for publication.

Firstly, extensive english editing is still needed. I would suggest making use of the mdpi English editing service.

Secondly, each statement should be supported by an adequate No of recent and scientifically sound references. This does not happen in the manuscript submitted. Please kindly upgrade both the size and the "distribution" of your reference library/

Author Response

After following the comments of both myself and reviewer no3 and revising the manuscript accordingly, you have managed to present a well improved version of the original draft.

R: We thank the reviewer for the positive comment. 

However, there are still issues to be resolved before the paper is considered for publication.

Firstly, extensive english editing is still needed. I would suggest making use of the mdpi English editing service.

R: thank you. We performed extensive English review of the manuscript. 

Secondly, each statement should be supported by an adequate No of recent and scientifically sound references. This does not happen in the manuscript submitted. Please kindly upgrade both the size and the "distribution" of your reference library/

R: We thank the reviewr for the suggestion. We increased the number of reference especially in the introducion and "comparison with other immunotherapies" parts. Style of the references was optimized too. 

Reviewer 3 Report

I recommend the manuscript for publication.

Author Response

We thank the reviewer for the positive comment. We revised also English language and style. 

Round 3

Reviewer 2 Report

Dear Authors, Dear Editors,

The revised manuscript is a major improvement to the older versions.

My final suggestion before the manuscript is published would be for the authors to extend the Figure legend No 2 (Overview of immune-based therapeutic approaches under investigation) in order to better describe what is shown in the figure.

Author Response

We thank the reviewer for the positive comment and suggestion! We extended the legend of Figure 2.